# Recent Advances in Copper-Based Organic Complexes and Nanoparticles for Tumor Theranostics

**DOI:** 10.3390/molecules27207066

**Published:** 2022-10-19

**Authors:** Sergey Tsymbal, Ge Li, Nikol Agadzhanian, Yuhao Sun, Jiazhennan Zhang, Marina Dukhinova, Viacheslav Fedorov, Maxim Shevtsov

**Affiliations:** 1International Institute of Solution Chemistry of Advanced Materials and Technologies, ITMO University, 197101 Saint Petersburg, Russia; 2Cancer Center & Department of Breast and Thyroid Surgery, Xiang’an Hospital of Xiamen University, School of Medicine, Xiamen University, 2000 Xiang’an Road East, Xiamen 361101, China; 3Xiamen Key Laboratory for Endocrine-Related Cancer Precision Medicine, Xiang’an Hospital of Xiamen University, Xiamen 361101, China; 4Guangxi University of Chinese Medicine, Nanning 530200, China; 5Day-Care Department, Xinjiang Medical University, Urumqi 830011, China; 6Laboratory of Biomedical Nanotechnologies, Institute of Cytology of the Russian Academy of Sciences, 194064 Saint Petersburg, Russia; 7Personalized Medicine Centre, Almazov National Medical Research Centre, 2 Akkuratova Str., 197341 Saint Petersburg, Russia; 8Department of Radiation Oncology, Klinikum Rechts der Isar, Technical University of Munich, 81675 Munich, Germany

**Keywords:** copper, organic complexes, nanoparticles, tumor theranostics

## Abstract

Treatment of drug-resistant forms of cancer requires consideration of their hallmark features, such as abnormal cell death mechanisms or mutations in drug-responding molecular pathways. Malignant cells differ from their normal counterparts in numerous aspects, including copper metabolism. Intracellular copper levels are elevated in various cancer types, and this phenomenon could be employed for the development of novel oncotherapeutic approaches. Copper maintains the cell oxidation levels, regulates the protein activity and metabolism, and is involved in inflammation. Various copper-based compounds, such as nanoparticles or metal-based organic complexes, show specific activity against cancer cells according to preclinical studies. Herein, we summarize the major principles of copper metabolism in cancer cells and its potential in cancer theranostics.

## 1. Introduction

Copper is a transition metal that plays several important roles crucial for maintenance of cell homeostasis, regulation of cell growth and proliferation, and iron metabolism [1]. Various roles of copper are explained by its ability to act as either a recipient or a donor of electrons depending on the oxidation state: Cu^1+^ (cuprous ion) and Cu^2+^ (cupric ion). The oxidation state also affects the copper interaction with organic compounds. Thus, Cu^1+^ preferentially binds to the thiol group in cysteine or the thioether group in methionine, while Cu^2+^ exhibits a high affinity for the secondary carboxyl group in aspartic/glutamic acid or the imidazole nitrogen group in histidine. As a result, copper ions readily form complexes with biomolecules containing these amino acid residues. Copper atoms are involved in a functioning of a wide spectrum of proteins, such as copper/zinc superoxide dismutase (Cu/Zn SOD or SOD1) [2], cytochrome c oxidase (COX) [3], lysyl oxidase (LOX) [4], mitogen-activated protein kinase MEK1 [5], and cAMP-degrading phosphodiesterase PDE3B [6]. In these proteins, copper ions participate in diverse biochemical reactions (especially redox reactions) of donating or accepting of electrons and maintain specific protein structures by coordinating with the abovementioned groups.

Despite its important physiological role, free copper ions are able to damage DNA and protein molecules via generation of reactive oxygen species (ROS) and interaction with cysteine and methionine residues [7]. That is why each cell and whole organisms have distinct mechanisms for the regulation of copper absorbance, distribution, accumulation, and excretion. With the development and propagation of copper-based pharmaceuticals, it is crucial to consider these metabolic and regulatory pathways to improve biocompatibility and efficacy of such compounds. For now, only a small number of studies dedicated to the design of novel copper-containing compounds consider underlying molecular mechanisms of intracellular copper regulation. The present work aims to provide a holistic view of the problem to help researchers boost their work and realize rational approaches in drug development.

## 2. Copper Intake, Distribution, and Efflux in Normal and Tumor Cells

The major proteins involved in copper maintenance include: CTR1 (copper transport protein), which is responsible for copper intake either from the intestine or blood; metallochaperones and metallothioneins, including ceruloplasmin, which are responsible for metal sequestration, distribution in organisms, and transport to various proteins; ATP7A and ATP7B (ATP-ase copper transporter alpha) responsible for copper excretion via membrane efflux or Golgi apparatus [8]. All these proteins have cysteine- or methionine-rich domains responsible for the binding. A precise description of proteins involved in copper homeostasis and a comparison of copper metabolism in normal and cancer cells are given below.

As it has previously been mentioned, copper intracellular metabolism is precisely regulated by specific protein machinery, which prevents the generation of free copper ions in the cytoplasm or extracellular space and ion-mediated toxicity (Figure 1). CTR1 is a major protein responsible for copper uptake in eukaryotes. CTR1 transporter acts as a pump that facilitates copper import without ATP consumption [9]. The rate of the copper intracellular transport depends on the copper concentration, the presence of other ions (Fe^3+^, Zn^2+^, Ag^+^) and organic compounds (e.g., ascorbate), cell type, and pH. The structure of homotrimeric CTR1 protein contains methionine gates for selective bypass of monovalent copper ions exclusively. However, isoelectric silver ions can compete with copper decreasing its intracellular content [10]. As only monovalent copper can be transported by the CTR1 protein, bivalent copper should first be restored to the monovalent state. This process is facilitated by the reductase proteins, such as STEAP, which are also reported to be overexpressed in several types of cancers and involved in tumorigenesis [11].

After crossing the plasma membrane, copper ions are readily sequestrated by the numerous intracellular metallothioneins, metallochaperones, albumins, glutathione, and ceruloplasmin [12]. Some of these proteins can store the metal for further use, while others serve for intracellular transportation of copper. For example, metallochaperones transfer copper to the active centers of the certain proteins or buffer the metal for further use. Copper chaperon for superoxide dismutase (CCS) delivers copper to superoxide dismutase (SOD1) enzyme, which converts superoxide radical into hydrogen peroxide and oxygen [13,14]. COX17 is another metallochaperone responsible for copper transportation to COX, an important protein involved in oxidative phosphorylation [15].

Atox-1 is a transport protein that delivers copper to ATP7A and ATP7B, which are responsible for copper release into the blood or bile, respectively [16]. Additionally, ATOX1 can migrate to the cell nucleus and act as a transcription factor facilitating cell growth, proliferation, and migration. Another common localization of ATOX1 is in proximity to the plasma membrane, where copper can be transferred to the membrane-associated proteins, such as lysyl oxidase, which is involved in cell migration [16].

## 3. Copper Regulation in Cancer

Many proteins required for copper metabolism are known to be overexpressed or malfunctioned in cancer cell metabolism. The most known example is participation of these proteins in chemotherapeutic response to conventional drug cisplatin. There is much evidence about CTR1 involvement in the transportation of cisplatin [17,18]. Meta-analysis of gene expression in various cancer types revealed that the reduced expression of the CTR1 gene is associated with the development of cisplatin resistance [19]. The knockout of CTR1 and DMT1 (divalent metal transporter 1) in human H1299 non-small cell lung cancer cells leads to pronounced cisplatin resistance. Moreover, the CTR1 loss decreases expression of COMMD1, XIAP, and NF-κB, which have a distinct influence on the intracellular homeostasis and signaling [20]. Several works of various research groups also proved a hypothesis about involvement of CTR1 and ATOX1 in cisplatin transport and sequestration [21,22,23]. However, another study on HEK-293T cells provided evidence about the modest participation of copper-binding proteins (i.e., CTR1, CTR2, ATOX1, and CCS) in cisplatin uptake and distribution [24].

At the same time, a connection between high ATOX1 expression level and survival rate in primary tumor biopsies has been found. Analysis of transcription profiling of 1904 breast cancer patients on METABRIC data set suggests that overexpression of Atox1 may serve as a marker for breast cancer prognosis [25] but only in the hormone receptor-positive tumors. Considering copper involvement in the functioning of the LOX protein [26] which is responsible for cell migration, ATOX1 may facilitate the function of LOX enhancing tumor ability for metastasis [27]. Moreover, Atox1 is also involved in transcription regulation of several genes, as was mentioned earlier. First, upon copper binding ATOX1 can migrate to the cell nucleus and bind the cis element of the cyclin D1 promoter, thus stimulating cell growth and proliferation [21]. Furthermore, a more complex interplay between ATOX1 and p53 has been found [28]. Authors observed increased copper amounts in cell nuclei for HCT116 p53^+/+^ cells compared to p53^−/−^ cells. These facts suggest that Atox1 may play a significant role in cell signaling and regulation of gene expression which should be determined in future studies.

Cytochrome c oxidase copper chaperone (COX17) is also involved in cancer. Inhibiting COX17 in acute leukemia cells results in decreased adenosylhomocysteinase activity leading to disruption of DNA methylation and changes in cell epigenetics [29]. The link between COX17 and cisplatin distribution to mitochondria has been found [30]. The involvement of copper-binding proteins in cisplatin uptake and distribution is probably connected to the similarities in binding affinity of platinum and copper ions. Moreover, glutathione (GSH) seems to attenuate this effect. It was found that 90% of cisplatin bound to GSH is readily transferred to COX17 [31]. This suggests probable involvement of thiol-containing molecules and not only proteins in intracellular cisplatin distribution. It would be interesting to investigate the effects of combining treatment with cisplatin and thiols or cisplatin-thiol complexes or nanostructures. COX17 was also studied as a prognostic marker for prediction of tamoxifen resistance in breast cancer patients [32]. The authors reported that this protein could be employed as a predictive marker for tumor recurrence and metastasis. These features are also observed for COX5B which is a subunit of COX itself [33]. This correlates with the prognostic value of ATOX1 which was found to possess similar properties in the breast cancer. Another COX nuclear-encoded subunit, COX4, is also shown to be a valuable prognostic and therapeutic marker for medullary thyroid cancer treatment [34]. The role of the COX protein in cancer development and progression as well as its influence on altered signaling and metabolic pathways needs to be further explored.

CCS, a protein involved in copper delivery to SOD1, is also involved in tumorigenesis. SOD1 could serve as a prognostic marker which contributes to worsened prognosis and higher risk of gastric [35] and prostate [36] cancer. Another study indicates SOD1 involvement in cell proliferation and metastasis in non-small cell lung cancer [37]. At the same time, knockdown of CCS leads to decreased cell proliferation and migration of MDA-MB-231 cells but does not affect the MCF-7 cell line [38]. In addition, the MAPK/ERK pathway was inhibited upon loss of CCS activity in MDA-MB-231 cells which also correlated to the increased ROS formation. Inhibition of CCS and Atox1 with specifically designed small molecules is a promising treatment strategy with reduced side effects [39]. The expression of CCS was found to be decreased in human hepatocellular carcinoma (HCC) which is distinct from breast cancer [40]. Despite a statistical significance not being achieved, the study concluded that a low expression level of CCS is a negative prognostic marker for HCC patients. Presumably, copper trafficking in various tissues could be different, as well as the involvement of copper-binding proteins in cancer development, progression, and metastasis. This provides a foundation for further investigation on a wide panel of cancer cell lines.

Copper efflux proteins, ATP7A and ATP7B, are also involved in cancer progression. ATP7A correlates with a poor survival rate and is overexpressed in several tumor types, such as breast, lung, prostate, ovarian, and colon cancer [41]. Another study shows that ATP7A is associated with cisplatin resistance in ovarian cancer and influence effectiveness of treatment with tetrathiomolybdate, which inhibits ATP7A activity [42]. Decreased sequestration of platinum leads to its accumulation in the cell nucleus with subsequent DNA damage. Moreover, the application of tetrathiomolybdate can also result in Ctr1 high expression increasing cisplatin uptake that may be used as a solution for treatment of drug resistance tumors [43]. Another study suggests a greater impact from inhibiting ATP7B compared to ATP7A [44]. A detailed analysis of the ATP7A and ATP7B roles in ovarian cancer are discussed in the review [45]. A study in the breast cancer model reveals the opposite effects of ATP7A and ATP7B in contribution to the cisplatin resistance [46]. ATP7A seems to be more involved in this process, whereas the analysis of ATP7B did not reach statistical significance. To summarize, the above-mentioned ATP7A and ATP7B influence the cisplatin efflux leading to decreased effectiveness of this drug; however, the precise role of each protein should be determined for distinct types of cancer.

Copper takes an active part in the proangiogenic pathways via several mechanisms. First, copper stimulates endothelial cells proliferation and migration. Next, copper is involved in the expression of certain proangiogenic factors (for example, vascular endothelial growth factor VEGF) [47], particularly as a response to hypoxia-inducible factor (HIF-1) signaling [48]. When elevated, copper becomes toxic and may induce side effects leading to genetic disorders (e.g., Wilson’s disease) and various types of oncological diseases. However, the exact molecular mechanisms underlying the connection between excessive copper levels and malignant cells are still unknown. It can only be hypothesized, particularly in the early stages, after considering the role copper plays in tumor angiogenesis. Malignant tissues have higher Cu accumulation levels, thus increasing the expression of human copper transporter (hCTR1). hCTR1 regulates the activation of cell-signaling pathways in embryogenesis, which leads to the development and progression of cancers [49].

The above-mentioned impact of copper ions and copper-binding proteins on cell growth, migration, and metabolism suggests that cancer cells require high copper levels to facilitate cell survival and disease progression. Indeed, tumor tissues are enriched with copper suggesting that this metal is one of the diagnostic tools for various oncological disorders [50]. Moreover, copper or copper-binding proteins are essential for the function of important signaling pathways, such as BRAF [51], NF-kB [52], MAPK [53], and EGFR/Src/VEGF [54]. Hence, the significant role of copper in cancer appearance and progression is starting to emerge in front of researchers. The accumulated data uncover the possibility to improve the efficiency of diagnostic approaches and increase treatment efficacy.

## 4. Therapeutic Effects of Copper-Based Compounds and Nanocarriers

The disparity in tumor cell and normal cell responses to copper have paved the way for copper complexes to evolve as anticancer agents. Copper-based compounds nowadays are receiving attention due to their target-specific therapeutic properties. Copper compounds influence the activities of several crucial cell organelles, such as the mitochondria and endoplasmic reticulum, leading to the loss of their functions and eventually resulting in cell death (Figure 2).

Nowadays, the increasing number of metal-based compounds and nanoparticles are being investigated due to their promising potential in theranostics, and various iron, zinc, copper-based and other agents are under development and testing for these purposes. For example, superparamagnetic iron oxide nanoparticles (SPIONs) are being actively used as a contrast agent for MRI procedures and in therapy. Currently, there are several running translational studies which explore SPIONs’ toxicity and biomedical applications, and ferumoxytol was FDA-approved for clinics [55]. Copper is also attracting the attention of researchers as a possible component for nanocompounds for theranostics and drug delivery. For example, copper is used in PET scanning as a radiotracer agent in cancer diagnostics, and 64CuCl_2_ has successfully passed clinical studies demonstrating its diagnostic potential [56]. Several studies successfully implemented copper for efficient bone regeneration [57] and anti-inflammatory therapy [58]. Copper-based nanoparticles also found their place in chemodynamic [59] and photothermal therapy [60].

The radiotracer biodistribution has shown that the liver has the highest uptake, followed by the intestine and pancreas, with urinary excretion being insignificant. It is the first biodistribution and radiation dosimetry trial with healthy volunteers. The estimated absorbance and effective doses were higher than the ones from another report with participants suffering from prostate cancer. The measurement methodology and assumptions used in dose calculation as well as the difference between the biodistribution in cancer patients and healthy volunteers are the main reasons for that disparity [61]. An interesting combination of SPIONs and Cu (II) ions were used as a cell labeling MRI/PET agent. Contrast agents showed good cellular uptake and cell-labeling ability [62]. Furthermore, gold nanoparticles alloyed with copper-64 demonstrate higher sensitivity and stability compared to non-modified gold nanoparticles [63]. Thus, copper presence could improve the effectiveness of the iron or gold nanoparticles, which opened new opportunities for further research in the field of cancer imaging. However, the major limitation and risk factor for wide implication of copper is toxicity of copper ions for cells [64].

Extrinsic and mitochondrial pathways of apoptosis are important in the control of tumor development and could be exploited for therapy [65]. The anticancer properties of Schiff base copper (II) complexes are well-studied and known in the scientific community. For instance, [Cu(sal-5-met-L-glu)(H_2_O)].H_2_O, [Cu(ethanol)2(imidazole)4][Cu2(sal-D, L-glu)2(imidazole)2] and [Cu(sal-D,L-glu)(2-methylimidazole)] complexes activate the intrinsic pathway, while [Cu2(sal-D, L-glu)2(isoquinoline)2]·2C_2_H_5_OH initiates the extrinsic pathway in human HT-29 colon carcinoma cells, respectively. All these complexes also induce a cytotoxic effect on the HT-20 cell line, and as a result, prove that they might become potential anticancer agents [66]. Structural formulas of the complexes can be found in recent publications [67,68,69]. Another study shows that accumulation of copper ions inside the cells leads to oxidative stress and apoptosis [70]. Moreover, the usage of 2,2′-dithiodipyridine strongly enhances this effect which is bound to its ability to transport copper through the plasma membrane.

Topoisomerases play an essential role in DNA replication and are relevant in cancer research as a target for novel therapies. There are currently several drugs approved by the FDA targeting topoisomerases (e.g., irinotecan, etoposide, etc.). Thiosemicarbazones are a group of complexes proved to have anticancer activity. “Triapine” (thiosemicarbazone) has been successfully tested for uterine cervix and vaginal cancers in clinical trials phase I and II and is presently under clinical trials phase III [71]. Thiosemicarbazones copper (II) complex [Cu(PyCT4BrPh)Cl] was investigated and demonstrated a cytotoxic effect on a leukemia cell line (THP-1) and human breast cancer cell line (MCF-7). It had stronger topoisomerase inhibitor activity and generally more impact on these cell lines than its analogue without copper, which proves how transition metals can increase the effectiveness of the known compound [72].

Copper complexes are shown to influence the endoplasmic reticulum leading to immunogenic cell death in breast cancer stem cells [73,74,75]. In a recent study, cuprous oxide nanoparticles affect calcium transport leading to its accumulation in intracellular space resulting in oxidative stress, activation of caspases, and apoptosis. Copper complexes are also able to inhibit proteasome function [76]. Other structures allow G-quadruplex telomeric DNA reduction [77]. These effects lead to disturbances in cell cycle, activation of apoptotic pathways, and cancer cell death. One article reports copper complexes are able to accumulate inside mitochondria leading to cytotoxicity by damaging mtDNA [78]. A great variety of induced effects allows copper compounds to be used for various applications in a precisely determined manner of action.

## 5. Copper Nanoparticles for Cancer Imaging and Drug Delivery

Due to the recent developments in imaging technologies and biology, molecular imaging provides not only the possibility to visualize the tumor, but also to assess the expression and activity of specific molecules (e.g., protein kinases, enzymes, proteases, etc.) and various processes (including metastasis, tumor cell apoptotic death, angiogenesis, etc.) involved in cancer progression, response to therapy, and recurrence [79]. Furthermore, molecular imaging based on CuS NPs enables repetitive assessment of particles biodistribution and biokinetic properties employing positron emission tomography (PET) and photoacoustic imaging (PAI) [80,81].

Photoacoustic (PA) imaging, developed rapidly in the recent decade, represents a noninvasive biomedical imaging method which can be employed for visualization of deeply located tissues tumors, analysis of vasculature [82], or evaluation of neoangiogenesis [83]. Upon the in vivo absorbance of a short-pulse laser by various molecules (e.g., water, melanin, RNA, DNA, hemoglobin, cytochromes, lipids, etc.) ultrasonic signals are generated via the mechanism of photothermal conversion [84,85,86]. Up-to-date gold nanostructures (GNPs) were widely applied as contrast agents for photoacoustic imaging [87]. However, GNPs were reported to have several limitations as contrast agents, including dependence of optical properties on shape, geometry, and size of particles as well as their susceptibility to tumor microenvironmental factors. On the contrary, compared to the maximum absorption between 560 and 840 nm of GNPs, the absorption of copper nanoparticles could be tuned to peak at wavelengths greater than 900 nm, thus providing the improved sensitivity in the NIR region (i.e., stronger PA signal, higher signal-to-noise ratio, greater field-of-view) [88]. Indeed, in the study by Zhou [89] et al., it was shown that polyethylene glycol (PEG)-coated copper(II) sulfide nanoparticles (PEG-CuS NPs) (peak absorption of 1064 nm) could be successfully employed both as a contrast agent for in vivo imaging of 4T1 breast tumor vasculature and as a mediator for photothermolysis of cancer cells. However, due to the intrinsic dipole–dipole interactions among Cu-based particles, synthesis of size-tunable, biocompatible, and colloidally stable suspension of particles remains a challenge. To overcome this problem Ding [90] et al. proposed the aqueous synthesis of PEGylated copper sulfide particles with controllable size between 3 and 7 nm. Subsequent preclinical studies demonstrated that particles, particularly of less than 5 nm, had a higher tumor-imaging potential. Another approach could be based on application of tumor microenvironment-sensitive nanoparticles as was proposed in the work of Wang et al. [91]. The authors developed iron-copper co-doped polyaniline nanoparticles (Fe-Cu@PANI) which upon glutathione (GSH) redox reaction could shift in the absorption spectrum from the visible to the NIR. The etching of Fe-Cu@PANI resulted both in photoacoustic imaging of tumors and efficient photothermal therapy. In recent research by Bindra [92] et al., the authors synthesized a self-assembled nanosystem (SCP-CS) which consisted of a semiconducting polymer (SCP) and encapsulated ultrasmall CuS (CS) nanoparticles. This nanosystem demonstrated not only an improved PA-imaging ability but also significant tumor growth inhibition due to the enhanced production of ROS.

In PET apart from traditionally employed positron emitters [64Cu]-based NPs were also shown as an efficient radiotracer for tumor diagnostics [93,94]. Thus, Zhou [94] et al. in the U87 human glioblastoma xenograft model demonstrated that a novel class of chelator-free [64Cu]CuS nanoparticles (NPs) (PEG-[64Cu]CuS NPs) could effectively target the tumor cells providing a potential for image-guided PTA therapy. In a more recent study, more complex indium- and copper-based metal-phenolic nanoparticles (MPNs) (labeled with 111In and 64Cu) were proposed for in vivo multimodal PET/SPECT/CT imaging [95].

Among other applications of Cu-based NPs is their use as a chemotherapeutic drug delivery system. Recently, Zhang [96] et al. proposed hybrid hollow mesoporous organosilica nanoparticles (HMONs) that consisted of ultrasmall photothermal CuS particles and disulfiram (DSF). Upon near-infrared (NIR) irradiation, released Cu^2+^ ions from nanoparticles converted the nontoxic DSF into a highly cytotoxic diethyldithiocarbamate (DTC)-copper complex that inhibited tumor growth. In another study, thermo-responsive copper sulfide (CuS) was employed to deliver CRISPR-Cas9 ribonucleoprotein (RNP) and doxorubicin for tumor combination therapy consisting of chemotherapy, gene therapy, and photothermal therapy [97].

## 6. Clinical Application of Copper-Based Nanoparticles in Oncology

Although some breakthroughs have been made in the treatment of malignant tumors [98,99], therapies, such as chemotherapy and radiotherapy, have become the most commonly used clinical treatments for tumors. However, the recurrence rate, drug resistance, quality of life, and other issues of cancer patients are still a global challenge [100]. In recent years, nanomaterials can effectively deliver drugs to specific targets, protect blood circulation drugs from endogenous enzymes, extend the half-life of drugs, and have shown great potential in tumor treatment [101,102].

Breast cancer (BC) is the second most common female cancer in the world, second only to lung cancer [99]. Studies have shown that copper-based nanomaterials have broad application prospects in the treatment of BC. For example, Ahamed et al. [103] found that copper ferrite (CuFe_2_O_4_) nanoparticles (NPs) added to the culture of human breast cancer MCF-7 cells can cause intracellular oxidation stress response, exerting anti-cancer effects, specifically manifested in the production of ROS and the consumption of glutathione (GSH) (Figure 3). Furthermore, Rajagopal et al. [104] found that copper nanoparticles (Wt-CuNPs) have obvious cytotoxic effects on MCF-7 cells. The specific mechanism is mainly due to the release of copper ions from the nanoparticles and the binding of copper ions to tumor cell DNA, causing DNA damage and the resulting apoptotic cell death.

Copper-based nanomaterials have also achieved good results in the treatment of esophageal cancer. Wang et al. [105] covered the silica coating on the Cu9S5 nanoparticles to form Cu9S5@MS core-shell nanostructures and added Cu9S5@MS core-shell nanostructures to human esophageal squamous carcinoma Eca109 and TE8 cells. After co-cultivation and treatment with NIR, it was found that Cu9S5@MS + NIR performs active anticancer activity against the EC109 and TE8 cancer cell lines by cell cycle arrest (Figure 4).

Furthermore, Xu et al. [106] optimized the concentration of disulfiram and Cu^2+^ ion for inhibiting esophageal cancer cells and loaded them in hyaluronic acid (HA)/polyethyleneimine (PEI) nanoparticles with specific scales to obtain NP-HPDCu^2+^ nanoparticles to improve the effectiveness and targeting of the drug. In vitro experiments proved that NP-HPDCu^2+^ nanomaterials can significantly promote the occurrence of Eca109 cell apoptosis and inhibit the migration and invasion of Eca109 (Figure 5). At the same time, the nude mouse tumor model proves that NP-HPDCu^2+^ nanomaterials can reduce the tumor volume and keep the weight of nude mice stable. The results of tumor tissue immunohistochemistry, immunofluorescence staining, and western blotting also showed that NP-HPDCu^2+^ nanomaterials can promote apoptosis and inhibit proliferation of esophageal squamous cell carcinoma.

Lung cancer is the malignant tumor with the highest mortality rate in the world, and non-small cell lung cancer is the most common pathological type in clinic [107,108]. Some researchers have found that copper-based nanomaterials have shown great potential in the treatment of NSCLC. Naatz et al. [109] constructed a new type of nanomaterial, Fe-doped CuO nanomaterial, which can use doped Fe to control the dissolution kinetics of copper-based nanomaterials. Using mouse lung squamous cell KLN-205 to construct a tumor-bearing nude mouse model by regulating the release of Cu^2+^, the local long-term drug concentration can be maintained, and the occurrence of drug resistance can be reduced. Additionally, these particles can also trigger a systemic anti-cancer immune response, promote the generation of ROS, and increase the rate of tumor cell death, which shows that CuO nanomaterials also have broad prospects for anti-cancer applications (Figure 6). In addition, Kalaiarasi et al. [110] reported that in A549 cells, the anti-cancer effect of CuO copper-based nanomaterials is related to the inhibition of histone deacetylase (HDACs) expression. Specifically, CuO copper-based nanomaterials have a strong inhibitory effect on different types of HDACs, can down-regulate the expression of oncogenes and up-regulate the expression of tumor suppressor genes, and induce apoptosis of cancer cells by activating the caspase cascade pathway to exert anti-cancer effects.

In recent years, with the continuous in-depth research of nanomaterials compared with traditional antitumor treatments, nanomaterials have been used in more and more clinical anticancer applications, showing great development potential [111]. For example, in our previous research, we found that some nanoparticles, such as superparamagnetic iron oxide nanoparticles (SPIONs), high-Z gold nanoparticles following intratumoral injection can provide a high local concentration of the agent, reduction of the particle clearance (i.e., renal or hepatic clearance) that increases the bioavailability of nanoparticles and has the effect of radiosensitizer in cancer radiotherapy, which can be used for long-term local anti-tumor therapy [112,113]. As an ideal anti-tumor drug candidate, copper-based nanomaterials have the following advantages: (i) compared with other metals, copper is cheap and rich in content [114], (ii) copper can induce reactive oxygen species (ROS)-mediated oxidative stress and promote tumor cell apoptosis [115,116,117], (iii) it has good biocompatibility, biodegradability, antibacterial properties, and selective cytotoxicity to cancer cells [118], and (iv) copper-based nanomaterials have less toxic effects on normal cells, fewer side effects, and are safer and more reliable [119]. Thus, copper-based nanomaterials have attracted more and more attention and have become the current research hotspot. At the same time, the emergence of copper-based nanomaterials has brought dawn to the treatment of various tumors [120].

However, copper-based nanomaterials also have limitations. For example, the production process of copper-based nanomaterials uses physical and chemical methods that are harmful to the environment and the human body [121]. Additionally, the instability and susceptibility to oxidation of copper-based nanomaterials under physiological conditions may also hinder its anti-tumor effect and reliability [122]. Furthermore, the biological safety of copper-based nanomaterials still requires further cell and molecular studies to avoid any impact on health, since Fahmy et al. [123] found that copper/copper oxide nanoparticles showed cytotoxicity to normal human lung WI-38 cells, resulting in the production of reactive oxygen species and DNA damage and inhibiting the growth and proliferation of WI-38 cells. The stability of copper nanoparticles is also one of the major concerns as copper tends to aggregate to the proteins, specifically cysteine and methionine residues. One work also found a dependency between stability and pH value [124]. However, using green synthesis, the authors successfully designed NPs which are mostly stable at various pH levels.

In short, copper-based nanomaterials are currently ideal anti-tumor drug candidates. With the continuous development of nanomaterials research, it will help provide better cancer treatment strategies in the future.

## 7. The Combination of Nanoparticles with Other Treatment Modalities

Based on the biological effects of copper and the physical and chemical properties of copper nanoparticles, their applications in the biomedical field mainly include externally triggered nanotherapies (photothermal therapy), drug delivery, antimicrobial applications, tissue regeneration, bioimaging, and bioeffects/biosafety. Therefore, it is reasonable to be expected that the construction of Cu-based biomaterials will have a unique integrated diagnosis and treatment function in clinical medicine. However, due to the complexity of tumors, such as the specific microenvironment and tumor metastasis, it is difficult to eradicate tumors completely through monotherapy alone. Therefore, the development of unique treatment modalities with multiple synergistic therapeutic performance has high prospects for improving therapeutic efficacy. Therefore, rational design of optimal drug combinations is important to achieve optimal synergistic therapeutic effects. Based on this, several unique multifunctional nanosystems involving copper have been constructed to jointly generate multiple nanotherapeutics [125].

Copper chalcogenides (Cu_2−x_E, E:S, Se, Te, 0 ≤ x ≤ 1) have been widely explored in photon-triggered disease therapy, such as photoacoustic imaging and photothermal hyperthermia. With stoichiometric ratios (Cu_2−x_S), deficient cuprous sulfide exhibits stoichiometric-dependent localized surface plasmon resonance (LSPR) absorption in the near-infrared range and photothermal conversion [126]. The integration of magnetic Fe_3_O_4_ nanoparticles exerted a magnetic targeting function to enhance tumor accumulation. Importantly, the photonic response of these Fe_3_O_4_@CuS composite nanoparticles in the second NIR biological window (1064 nm) achieves higher tissue penetration ability compared to the laser activation of the first NIR biological window. Thus, a higher tumor suppression rate was achieved with no further recurrence (808 nm). In addition to the photothermal conversion efficiency (25.7%) of hydrophilic plate-like Cu9S5 nanocrystals at 980 nm [127], the CuS superstructure was exemplified to respond to external 980 nm laser activation for photothermal conversion and subsequent cancer ablation [128]. The cysteine-coated CuS nanoparticles were also irradiated with a 980 nm laser with a high photothermal conversion efficiency of 38.0%, efficiently inhibiting tumor growth [129]. Furthermore, encapsulation of CuS nanoparticles into zeolite imidazole framework 8 (ZIF-8) resulted in NIR-induced dissociation of ZIF-8 to release loaded chemotherapeutics, aiming to achieve synergistic photothermal ablation and NIR-triggered chemotherapy [130]. Doping iron (Fe^3+^) can tune the vacancies of Cu_2−x_Se nanoparticles to control NIR absorption, which also enables these semiconductors to have MR-imaging properties [131].

To improve the photothermal conversion efficiency, Cu_2−x_S and Ag_2_S were integrated into one system by producing Cu-Ag2S/PVP nanoparticles with a high photothermal conversion efficiency of 58.2% under 808 nm laser irradiation, which is much higher than that of Cu_2−x_S/PVP nanoparticles (27.1%) [132]. The rational integration of plasmonic Au nanoparticles and plasmonic Cu_2−x_S semiconductors into one matrix can enhance the photothermal properties of Au or Cu_2−x_S components. The coupled LSPR properties of Au and Cu_2−x_S can be maximized by designing Au@Cu_2−x_S core/shell nanoparticles to enhance the PTT efficacy. Ji et al. synthesized Au@CuS nanoparticles and performed the following cation exchange between Cu^+^ and CdS shells, resulting in Au@Cu_2−x_S nanostructures [133], which can be formed as nanoparticles or nanorods. The corresponding photothermal conversion efficiencies are calculated to be 59% at 808 nm and 43% at 1064 nm, which rapidly increases the ambient temperature of the Au@Cu_2−x_S nanorod aqueous solution. In particular, the design of core/shell Au@Cu_2−x_S is more favorable compared to the simple mixture of Au nanorods and Cu_2−x_S nanoparticles for photothermal conversion. This core/shell design with improved photothermal performance also induced more HeLa cell death compared to the same concentration of Cu_2−x_S. The Au-Cu9S5 plasmonic hybrid nanosystem was established, which enhanced the LSPR of Cu9S5 through the coupling effect of LSPR based on the collective vibration of electrons and holes [134]. This Au-Cu9S5 hybrid nanosystem exhibits an absorption cross-section enhancement of 1.3 × 108 m^−1^ cm^−1^ and a high photothermal conduction efficiency of 37% for photothermal ablation of tumor tissue. According to the plasmonic coupling effect between core and shell, spherical Au@Cu_2−x_S, Au@Cu_2−x_S, and rod-shaped Au@Cu_2-x_S superparticles were synthesized for photothermal ablation of tumors (4T1 tumor model). It has X-ray-computed, tomography-imaging capabilities because of the presence of Au composition with a large atomic number and an X-ray attenuation coefficient (5.16 cm^−2^ kg^−1^) [135].

Photothermal therapy exposes materials with the photothermal conversion ability to near-infrared light. These materials can convert the absorbed light energy into thermal energy to kill tumors, showing excellent local tumor treatment effects, but they are less effective for metastatic tumors. The combination of photothermal therapy and radiotherapy in tumor treatment can achieve a synergistic effect. Thus, Zhou et al. [89] synthesized PEG-[64Cu]CuS NPs based on a single radioactive copper sulfide nanoparticle. The study demonstrated that inhibition of tumor growth was significantly high when both methods, radiotherapy and hyperthermia, were employed.

Photothermal therapy (PTT) mainly uses photothermal materials accumulated at the tumor site, which can convert the absorbed light energy into heat energy (above 45 °C) under near-infrared irradiation. Combining tumor photothermal therapy and immunotherapy could further improve the therapeutic potency of PTT [136]. Another approach could be based on the combination of PTT with chemotherapy. Thus, Wu et al. [137] demonstrated that encapsulation of CuS nanoparticles into the zeolite imidazole framework 8 (ZIF-8) resulted in NIR-induced dissociation of ZIF-8 to release loaded chemotherapeutics, which in turn provided synergistic photothermal ablation and NIR-triggered chemotherapy.

The tumor microenvironment is usually characterized by low pH [138], altered redox states [139], hypoxia [140], and expression of particular enzymes that could be employed for the development of stimuli-responsive nanoparticles. Based on the fact that the hydrogen sulfide (H_2_S)-producing enzyme of cystathionine-β-synthase (CBS) is upregulated in colon cancer, H_2_S concentrations in tumors reach approximately 0.3 to 3.4 mmol·L^−1^. Therefore, using this overexpressed endogenous H_2_S to convert cuprous oxide (Cu_2_O) to copper sulfide in situ can activate PA imaging and photothermal tumor ablation [141]. It is exemplified that the use of S-adenosyl-1-methionine (SAM) as an allosteric CBS activator accelerates the in situ reaction between H_2_S and Cu_2_O, resulting in significantly enhanced PA-imaging signal and photothermal effect. In contrast, the use of aminooxyacetic acid (AOAA) as a CBS inhibitor reduced the production of H_2_S and subsequently the conversion of Cu_2_O to copper sulfide, showing no significant PA signal and negligible temperature change in tumors. However, the photothermal conversion efficiency after high-dose copper sulfide conversion is low, and the ideal photon therapy effect cannot be obtained. To address this critical issue, based on the LSPR-coupling effect between noble metals and plasmonic semiconductors, Tao et al. constructed Au@Cu2O plasmonic hybrids to enhance in situ H_2_S-triggered post-conversion photothermal performance [142]. Similar to the conversion of Cu2O to Cu9S8, tumor-accumulated Au@Cu2O nanoparticles were also converted into Au@Cu9S8 nanoagents to achieve PA-enhanced contrast agents and photothermal tumor ablation by increasing tumor temperature. The LSPR-coupling effect induces nearly 2.1-fold stronger NIR absorption and 1.2-fold higher photothermal conversion efficiency, enabling the utilization of low nanoparticle doses with desirable therapeutic properties. These two paradigms provide another strategy for realizing photothermal hyperthermia involving copper-based nanoagents by in situ generation of copper-based nanoagents with unique photothermal properties. Cheng Y. et al. [143] took advantage of the ordered large-pore structure and easily chemically modified the property of DLMSNs, the copper sulfide (CuS) nanoparticles with high photothermal conversion efficiency. A homogenous cancer cell membrane was coated on the surfaces of these DLMSNs, followed by conjugation with the anti-PD-1 peptide. The thus-obtained AM@DLMSN@CuS/R848 was applied to holistically treat metastatic TNBC in vitro and in vivo. The data showed that AM@DLMSN@CuS/R848 had a high TNBC-targeting ability and induced efficient photothermal ablation on primary TNBC tumors under 980 nm laser irradiation. Tumor antigens thus generated and increasingly released R848 by response to the photothermal effect, combined with AUNP-12 detached from AM@DLMSN@CuS/R848 in the weakly acidic tumor microenvironment and synergistically exerted an anti-tumor effect, thus preventing TNBC recurrence and metastasis.

Table 1 summarizes the above information presenting major classes of therapeutics and some examples for detail consideration. The unique features of copper allow to create a wide spectrum of various nanostructures with great diversity of their applications.

## 8. Conclusions

Copper is an essential trace element in cell metabolism with distinct features. Participation of copper in oxidation–reduction reactions has an important impact on cell metabolism, survival, and growth. Free copper ions could exert a cytotoxic effect; however, most of the copper is bound to the enzymes, metallochaperones, and metallothioneins. These proteins, despite their direct function, could influence functionality of other proteins affecting cell signaling and gene expression, interfering in the anti-cancer chemotherapies. Recent studies demonstrate that copper-based nanocarriers due to their unique physio-chemical properties could be efficiently employed for tumor theranostics as a monotherapeutic approach or in combination with other treatment modalities. Constant development and modification of existing systems have great potential in clinic. Some limitations, which include ROS generation and free ion emergence, should be considered. However, an understanding of the underlying molecular regulation of copper intracellular distribution and metabolism will help to improve the current development of copper-based therapeutics and nanostructures for further efficient clinical application.

## Figures and Tables

**Figure 1 molecules-27-07066-f001:**
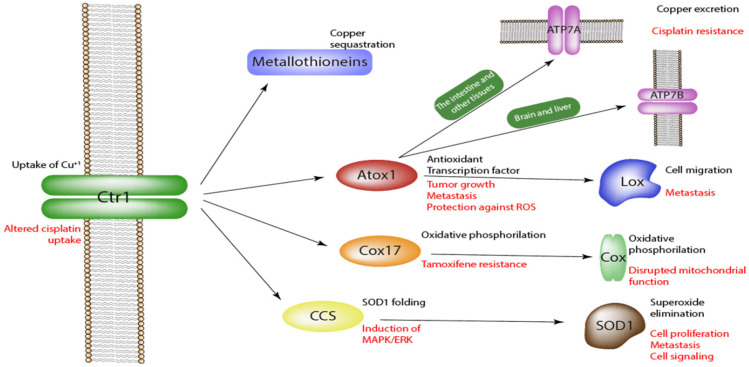
Proteins of copper metabolism. The arrows show how copper is metabolized. In general, CTR1 is responsible for copper intake; numerous metallothioneins and chaperones store the metal and deliver it to the active sites of functional proteins. The black font indicates the role of a protein in normal cells, and the red font indicates the protein function in cancer.

**Figure 2 molecules-27-07066-f002:**
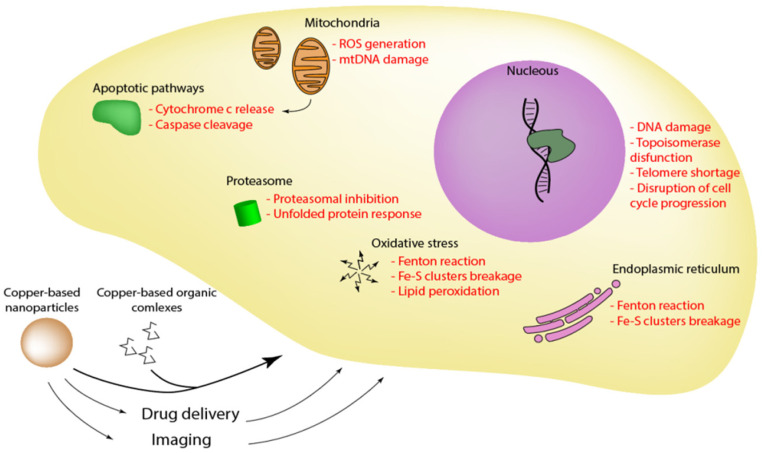
The main effect of copper nanoformulations on cell metabolism. The major impact of copper on cell metabolism is a result of ROS generation and DNA damage. Proteasome, endoplasmic reticulum, and mitochondria also suffer from copper excess.

**Figure 3 molecules-27-07066-f003:**
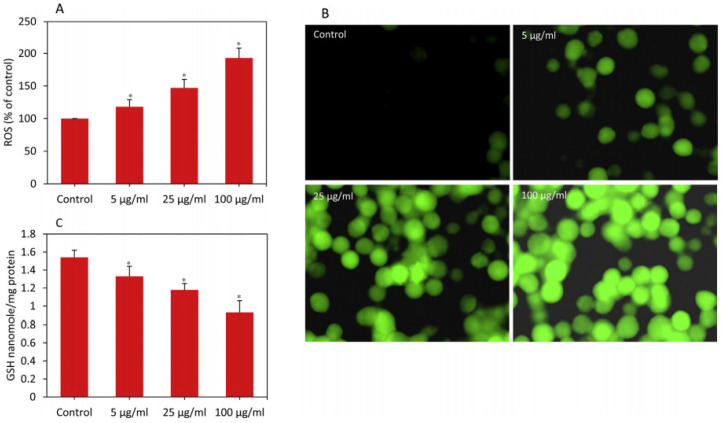
Copper ferrite NP-induced oxidative stress in MCF-7. Cells were exposed to copper ferrite NPs at the dosages of 0, 5, 25, and 100 g/mL for 24 h. At the end of exposure, ROS and GSH levels were determined, as described in materials and methods. (**A**) Percentage change in ROS level. (**B**) Fluorescence microscopy image of ROS generation. (**C**) GSH level. Data represented are mean ± SD of three identical experiments made in three replicates. * Significant difference as compared to control (*p* < 0.05).

**Figure 4 molecules-27-07066-f004:**
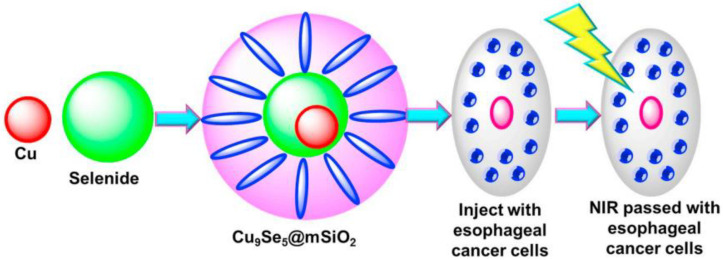
Portrayal of the Cu9S5@MS nanoparticles synthesis and application as a dual functional treatment stage for esophageal squamous carcinoma treatment.

**Figure 5 molecules-27-07066-f005:**
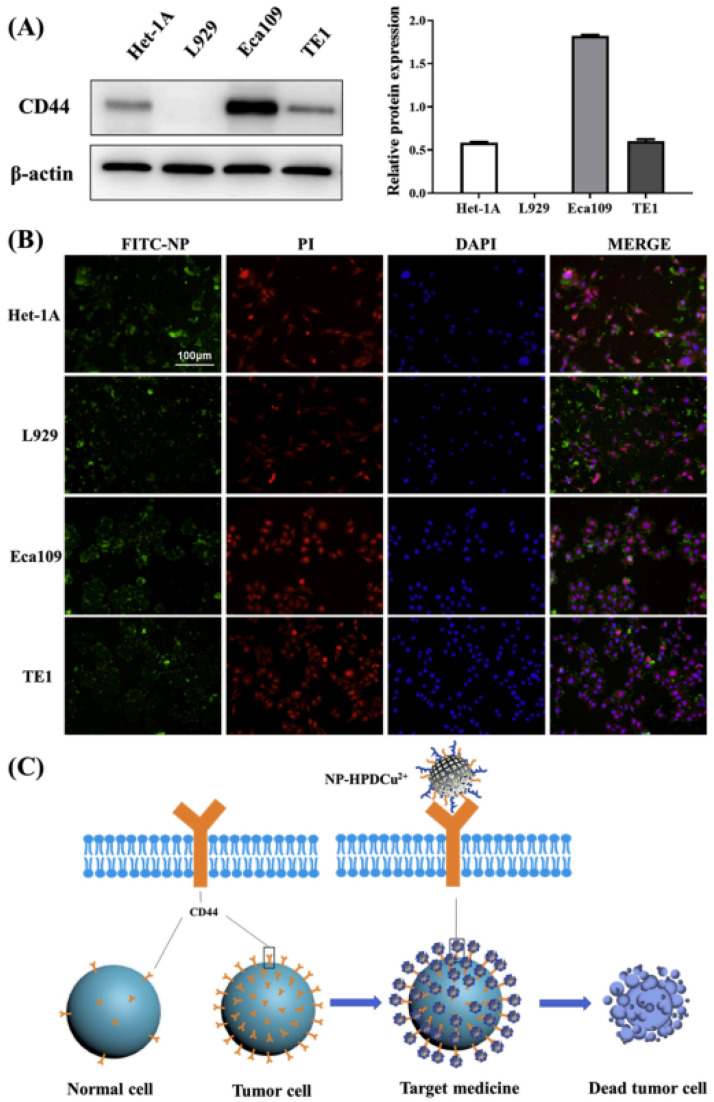
(**A**) Western blot of CD44 expressed on the Het-1A, L929, Eca109, and TE1 (mean ± SD, n = 3); (**B**) Fluorescence images of Het-1A, L929, Eca109, and TE1 stained with FITC-labeled NP-HPDCu^2+^ (FITC-NP, green color), PI (apoptosis marker, red color) and DAPI (nucleus marker, blue color); (**C**) mechanism diagram of targeted killing tumor cells by NP-HPDCu^2+^ nanoparticle.

**Figure 6 molecules-27-07066-f006:**
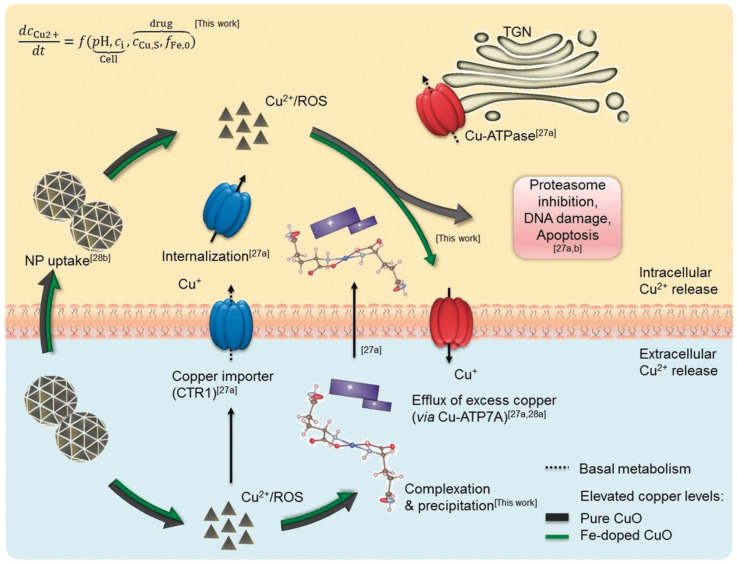
Copper homeostasis and regulatory mechanisms, including extra- and intracellular dissolution of pure and Fe-doped CuO NPs.

**Table 1 molecules-27-07066-t001:** Copper-based compounds and nanoparticles with various applications and mechanisms of action.

	Copper-Based Compound	Mechanism of Action
Diagnostic tool	64-CuCl_2_ [64]	Contrast agent in PET/MRI scanning
	Combination of SPIONs and Cu(II) [62]	
	Gold-copper alloyed NPs [63]	
Therapeutic agent	Schiff base copper (II) complexes [66]	Activation of extrinsic or intrinsic apoptotic pathways
	Copper-based nanoparticles [96,103]	Copper ions release, oxidative stress, DNA damage
	Thiosemicarbazones copper (II) complex [72]	Topoisomerase inhibition
	Polypyridyl-Schiff-base copper complex [74]	Targets endoplasmic reticulum leading to immunogenic cell death
	G-quadruplex-targeting copper complex [77]	Rapid reduction of telomeres in cancer cells
	Ferrocenyl terpyridine copper complexes [78]	Targets mitochondria, causes mtDNA damage
Combined approach	Copper chalcogenides [126]	Photothermal ablation and NIR-triggered chemotherapy
	Alloyed CuAg or CuAu NPs [132,133]	
	PEG-[64Cu]CuS NPs [94]	Combined radiotherapy and hyperthermia against metastatic tumor cells
	Copper-doped iron NPs [109,131]	Magnetic guidance and copper release with subsequent oxidative stress

## Data Availability

Not applicable.

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
