# Peer review of "Recent Advances in Copper-Based Organic Complexes and Nanoparticles for Tumor Theranostics"

_molecules, 2022, doi:10.3390/molecules27207066_

Round 1

Reviewer 1 Report

The manuscript titled, Recent advances in copper-based organic complexes and nanoparticles for tumor theranostics, is a comprehensive review of the current exploited copper-based nanostructures by using various approaches. Moreover, the authors also gave a brief overview of the regulation of copper in cancer and its importance in cancer therapeutics. However, I have some minor comments.

1.       The grammar and the overall language need a thorough check.

2.       Some recent works can also be included in the relevant sections. Few recent publications:

Passeri, G., Northcote-Smith, J. and Suntharalingam, K., 2022. Delivery of an immunogenic cell death-inducing copper complex to cancer stem cells using polymeric nanoparticles. RSC advances12(9), pp.5290-5299.

Lu, H., Xu, S., Ge, G., Guo, Z., Zhao, M. and Liu, Z., 2022. Boosting Chemodynamic Therapy by Tumor-Targeting and Cellular Redox Homeostasis-Disrupting Nanoparticles. ACS Applied Materials & Interfaces.

Fanizza, E., Mastrogiacomo, R., Pugliese, O., Guglielmelli, A., De Sio, L., Castaldo, R., Scavo, M.P., Giancaspro, M., Rizzi, F., Gentile, G. and Vischio, F., 2022. NIR-Absorbing Mesoporous Silica-Coated Copper Sulphide Nanostructures for Light-to-Thermal Energy Conversion. Nanomaterials12(15), p.2545.

Ashraf, S., Qadri, S., Akbar, S., Parray, A. and Haik, Y., 2022. Biogenesis of Exosomes Laden with Metallic Silver–Copper Nanoparticles Liaised by Wheat Germ Agglutinin for Targeted Delivery of Therapeutics to Breast Cancer. Advanced Biology, p.2200005.

Author Response

COMMENT: The manuscript titled, Recent advances in copper-based organic complexes and nanoparticles for tumor theranostics, is a comprehensive review of the current exploited copper-based nanostructures by using various approaches. Moreover, the authors also gave a brief overview of the regulation of copper in cancer and its importance in cancer therapeutics. However, I have some minor comments.

  1. The grammar and the overall language need a thorough check.
  2. Some recent works can also be included in the relevant sections. Few recent publications:

Passeri, G., Northcote-Smith, J. and Suntharalingam, K., 2022. Delivery of an immunogenic cell death-inducing copper complex to cancer stem cells using polymeric nanoparticles. RSC advances12(9), pp.5290-5299.

Lu, H., Xu, S., Ge, G., Guo, Z., Zhao, M. and Liu, Z., 2022. Boosting Chemodynamic Therapy by Tumor-Targeting and Cellular Redox Homeostasis-Disrupting Nanoparticles. ACS Applied Materials & Interfaces.

Fanizza, E., Mastrogiacomo, R., Pugliese, O., Guglielmelli, A., De Sio, L., Castaldo, R., Scavo, M.P., Giancaspro, M., Rizzi, F., Gentile, G. and Vischio, F., 2022. NIR-Absorbing Mesoporous Silica-Coated Copper Sulphide Nanostructures for Light-to-Thermal Energy Conversion. Nanomaterials12(15), p.2545.

Ashraf, S., Qadri, S., Akbar, S., Parray, A. and Haik, Y., 2022. Biogenesis of Exosomes Laden with Metallic Silver–Copper Nanoparticles Liaised by Wheat Germ Agglutinin for Targeted Delivery of Therapeutics to Breast Cancer. Advanced Biology, p.2200005.

ANSWER: Thank you for your consideration. We performed a thorough linguistic editing of the manuscript and included the suggested references.

Reviewer 2 Report

A review by Maxim Shevtsov et al. presents a comprehensive view of the nature of the behavior of copper particles (complexes) in cancer cells, which has implications for the treatment and diagnosis of cancer cells. The work is well structured and the material is presented logically and sequentially, allowing the reader to easily grasp the subject. Illustrative material is present in the review; however, from the point of view of organic chemistry, drawings with structural formulas of ligands or the discussed copper complexes are missing. This review can be recommended for publication in Molecules after revision. 

Remarks. It is necessary to correct indices in chemical formulas in the text (for example, H2S, etc.). Most important, in my opinion, is to add figures with structural formulas of ligands (for example, the mentioned Schiff bases).

Author Response

COMMENT: The work is well structured and the material is presented logically and sequentially, allowing the reader to easily grasp the subject. Illustrative material is present in the review; however, from the point of view of organic chemistry, drawings with structural formulas of ligands or the discussed copper complexes are missing. This review can be recommended for publication in Molecules after revision. 

Remarks. It is necessary to correct indices in chemical formulas in the text (for example, H2S, etc.). Most important, in my opinion, is to add figures with structural formulas of ligands (for example, the mentioned Schiff bases).

ANSWER: The authors are thankful to the reviewer for thorough revision. We included the additional links to the original chemical formulas of listed compounds, in particular lines 255 and 256. Names of mentioned organic complexes were corrected.

Reviewer 3 Report

The paper entitled “Recent advances in copper-based organic complexes and nanoparticles for tumor theranostics” prepared by Sergey et e summarize the major principles of copper metabolism in cancer cells and its potential in cancer theranostics. The work is interesting, By the way, I recommend a major revision and it is needed to address the following issue:

-what is the main novelty of this work? There are plenty of similar papers

- There are some spelling errors and logic problems in the text that need attention. Moreover, the typos in the manuscript need to be double-checked. For example, Line 35

- The NPs stability and biocompatibility need to discuss by the authors

- It is better to provide a graphical abstract to present main goal of paper

- There are many studies investigating the importance of the topic , Please add these references to your introduction and discussion parts of the manuscript and compare and bold your study novelty: https://doi.org/10.1002/adfm.202202470, https://doi.org/10.1016/j.compositesb.2022.109855, https://doi.org/10.1021/acs.orglett.9b01230, https://doi.org/10.1016/j.mtchem.2022.101131

- what is the suggestion of this study for future works?

Author Response

COMMENT 1: The paper entitled “Recent advances in copper-based organic complexes and nanoparticles for tumor theranostics” prepared by Sergey et e summarize the major principles of copper metabolism in cancer cells and its potential in cancer theranostics. The work is interesting, By the way, I recommend a major revision and it is needed to address the following issue:

-what is the main novelty of this work? There are plenty of similar papers.

ANSWER1: The main novelty of the study is an in-depth analysis of molecular mechanisms of intracellular distribution and function of copper which are critical to consider for improvement of current cancer diagnosis and treatment. The Introduction and Discussion sections were improved to emphasize these aspects.

COMMENT 2: There are some spelling errors and logic problems in the text that need attention. Moreover, the typos in the manuscript need to be double-check

ANSWER 2: We have edited the text and removed typos and errors.

COMMENT 3: The NPs stability and biocompatibility need to discuss by the authors.

ANSWER 3: Thank you for the valuable comment. We included the paragraph about biocompatibility and stability of nanoparticles (lines 428 – 442).

COMMENT 4: It is better to provide a graphical abstract to present main goal of paper.

ANSWER 4: We included a graphical abstract.

COMMENT 5: There are many studies investigating the importance of the topic , Please add these references to your introduction and discussion parts of the manuscript and compare and bold your study novelty: https://doi.org/10.1002/adfm.202202470, https://doi.org/10.1016/j.compositesb.2022.109855, https://doi.org/10.1021/acs.orglett.9b01230, https://doi.org/10.1016/j.mtchem.2022.101131

ANSWER 5: Thank you for suggesting new materials. We included the references into the text to corresponding parts.

COMMENT 6: what is the suggestion of this study for future works?

ANSWER 6: We added lines 576-579 in which the prospects for future research are stated.

Round 2

Reviewer 3 Report

The paper can be accepted